# Effects of Long-Term Bottle Storage on Red and Rosé Wines Sealed with Different Types of Closures

**DOI:** 10.3390/foods10122918

**Published:** 2021-11-25

**Authors:** Prudence Fleur Tchouakeu Betnga, Edoardo Longo, Vakarė Merkytė, Amanda Dupas de Matos, Fabrizio Rossetti, Emanuele Boselli

**Affiliations:** 1Faculty of Science and Technology, Free University of Bozen-Bolzano, Piazza Università 1, 39100 Bolzano, Italy; prudencefleur.tchouakeubetnga@natec.unibz.it (P.F.T.B.); edoardo.longo@unibz.it (E.L.); vakare.merkyte2@unibz.it (V.M.); 2Oenolab, NOI Techpark Alto Adige/Südtirol, Via A. Volta 13B, 39100 Bolzano, Italy; 3Food Experience and Sensory Testing (Feast) Lab, Massey University, Palmerston North 4410, New Zealand; a.dupasdematos@massey.ac.nz; 4Riddet Institute, Massey University, Palmerston North 4410, New Zealand; 5Mérieux NutriSciences, Via Marradi 41, 59100 Prato, Italy; fabrizio.rossetti@mxns.com

**Keywords:** closures, dissolved oxygen content, sulfur dioxide, phenolic compounds, volatile profile, bottle storage, wine ageing

## Abstract

Volatile and non-volatile chemical profiles, free and total SO_2_ and dissolved oxygen content were studied in three red (Merlot, Lagrein red, St. Magdalener) and one rosé (Lagrein rosé) wine after 30 months of storage in bottles. Each wine was sealed with closures made of a ‘blend’ (B) of natural cork microgranules and polymers without glue and was compared with wines closed with other types of corks (C; a technical cork 1 + 1, or an agglomerated natural cork or a natural one-piece cork). Glutathionyl caftaric acid (GRP) was inversely correlated with total SO_2_ content and was higher in all three red wines closed with B compared to C, whereas epicatechin was higher in three wines closed with C compared to B. Three volatile compounds formed by fermentation (ethyl butanoate, isoamyl lactate, and octanol) were inversely correlated with both free and total SO_2_. In terms of their volatile profiles, ethyl octanoate and 2,3-butanediol were significantly higher in the Lagrein red wines closed with C closures, whereas no significant difference was observed in Merlot, Lagrein rosé and St. Magdalener wines. Small differences in some phenolic compounds due to the type of closure were found: GRP, syringic acid, (+)-catechin, and (−)-epicatechin differentiated the Merlot wines closed with B from the C closures. Protocatechuic acid and GRP levels differentiated the Lagrein red wines according to their closure type, whereas only (−)-epicatechin differentiated the Lagrein rosé wines. GRP, caffeic acid, (−)-epicatechin, and anthocyanin content differentiated the St. Magdalener wines according to their closure type. Even though St. Magdalener and Lagrein rosé closed with C could be distinguished from those closed with B by using the (sensory) triangle test (α = 0.05), these differences appeared to be relative as it did not include all the wines in a systematic manner.

## 1. Introduction

Some wines are intended to be stored for a short period of time before their consumption; in contrast, many premium wines are aged for several years or even decades [1]. Over the period of a bottle’s ageing, a wine gradually alters to reach maturity, at which point its organoleptic quality is maximally expressed; then, this period is followed by a phase of decline in quality [2]. The maturation of bottled wine is strongly linked to its composition at the time of bottling. In addition, storage conditions such as temperature and light exposure in the cellar are also relevant to the maturation of wine [3], but not all of them can influence the different aspects of wine quality in the same way (for example, aroma and color profiles). Wine ageing in bottles can be impacted by the quality of the closure used [4,5,6,7], which can drastically affect the commercial quality of the product, consumer preferences and, in general, the reputation of the winery. For example, a natural one-piece cork closure of low quality may be too permeable to air or may contract due to dehydration (due to a very low environmental relative humidity), leading to breakage of the closure, or to wine oxidation [8,9,10,11,12]. It is well known that the aroma of wine tends to develop because of numerous reactions. Wine can benefit from slight exposure to oxygen, as this allows color stabilization, favors the reduction of astringency and the development of specific aroma components [13]. This is particularly important for red wines, as they contain high levels of phenolic compounds, i.e., the main reactants with oxygen in wine [14]. Controlled oxidation can also prove beneficial to white wines, such as Riesling, which develops a golden (rather than brown) color and an aroma typical of aged wine [12,15,16]. A gradual maturation can also take place in an environment, such as in a bottle, which has a very low oxygen content [17,18]. In most scientific works, the maturation of wines exposed to oxygen after bottling has been evaluated, focusing on the relationship between the closure and wine exposure to oxygen [10,18,19]. Indeed, a high oxygen content or a low free sulfite content can lead to oxidative deterioration and consequently will result in some off-odors and/or off-flavors in the bottled wine [20,21].

Numerous studies have described the influence of closures during the maturation of wines after bottling, with the aim of establishing the optimal conditions for improving their organoleptic quality [5,19,22]. Oxygen exposure is the most important factor, along with the type of closure—although several other factors are reported as determinants during the post-bottling phase, such as light exposure [23,24], temperature fluctuations [25,26], vibrations [27,28], and humidity.

The effect of different closure materials on wine development has also been studied [6,28,29,30,31,32]. Godden et al. [29] evaluated the performance of different closures (natural cork, technical cork, screw cap, and synthetic closures) on bottled wines after 20 months of storage. The results showed that the highest retention of sulfites and ascorbic acid, as well as the slowest rate of browning, were recorded in wine bottles closed with screw caps. The direct interactions of the closure with the wine has been an important subject of research. González-Adrados et al. [6] conducted a study using two different closure materials (natural cork and 1 + 1 technical cork) with treated and non-treated surfaces to evaluate the interactions between the closure and the wine. Their results showed that most of the overall migration of non-volatile compounds from the closure to the wine was due to the natural components of the cork. Closure type (natural cork or 1 + 1 technical cork closures) and contact time accounted for the greatest variability, while surface treatment increased overall migration and decreased liquid absorption.

The first part of the present research [33] provided details on the maturation of four wines (one rosé and three reds from Alto Adige/South Tyrol, Italy) during up to 12 months of storage in bottles sealed with different closures.

In this study, the profiles of volatile and non-volatile compounds, free and total SO_2_ content, and dissolved oxygen content of the same wines were analyzed after 30 months of storage in bottles, with the goal of evaluating how quality was affected by the different types of closures over a much longer period of time than previously studied. In addition, a sensory discrimination test was applied to assess the differences between samples closed with two different closures (‘conventional’ *vs* ‘blend’). The results obtained broadened the knowledge of closure–wine interactions after a long period of storage in the bottle.

## 2. Materials and Methods

### 2.1. Wines and Closures

Four wines (Merlot, Lagrein red, Lagrein rosé and St. Magdalener) were provided by a local winery (Kellerei Bozen, Bolzano, Italy). The grapes were harvested in 2016 from vineyards located in the same area. Each of the four types of wine was made using the winery’s standard winemaking protocol for that particular wine; the wines were then bottled in the following year. The only difference between the bottles of each different wine was the type of closure used. The different closures were: a ‘blend’ cork (a sanitized micro-granule blend of natural cork with polymers without the addition of glue to the mix) and ‘conventional’ closures (an agglomerated natural cork, a natural one-piece cork and a technical cork 1 + 1) from different suppliers, as reported in Table 1. A total number of 16 (0.75 L) bottles with 4 bottles of each type of wine were sampled. All the bottles were stored horizontally at a cellar temperature (about 16 °C, 50% relative humidity) for most of the storage time. The bottles were opened after 30 months of storage to characterize their volatile and phenolic profiles, as well as their dissolved oxygen, free and total SO_2_ content and to perform the triangle test. At the time of analysis, all the bottles were stored in the laboratory for a short time under a constant controlled temperature (23 °C). A total of four bottles of each type of wine (two replicate samples for each type of closure) were analyzed.

### 2.2. Analysis of Volatile Compounds

Volatile compounds were sampled by head-space solid phase microextraction (HS-SPME), according to the operating parameters described by Rossetti et al. [33]. Then, they were determined by gas chromatography–mass spectrometry (GC–MS) with slight modifications. Briefly, 1 g NaCl was introduced into a vial (20 mL) containing 10 mL of wine; the vial was then closed with a screw cap fitted with a perforable elastomeric septum and was placed in a heating bath at 40 °C for 10 min. The vial was continuously stirred at 270 rpm to reach equilibrium. Afterwards, a SPME fiber (DVB/CAR/PDMS, 50/30 µm, 1 cm, Agilent, Santa Clara, CA, USA) was introduced into the sample headspace for 20 min under the same heating and mixing conditions.

For GC–MS analysis, a 7890A gas chromatograph was coupled to a 5975 mass spectrometer both from Agilent (Wilmington, DE, USA). Volatile compounds were thermally desorbed at 240 °C for 3.5 min and analyzed using a HP-5MS capillary column (0.25 µm/0.25 mm/30 m; Agilent, Wilmington, USA). The injector was set to splitless mode and the temperature program of the GC oven was as follows: 40 °C for 2.5 min, then increased to 180 °C at a rate of 3 °C/min and finally up to 230 °C at 10 °C/min. Helium, as the carrier gas, was set to a constant flow mode (0.7 mL/min). The electron ionization was set at an energy of 70 eV and the ion source temperature was set at 230 °C. The mass range of the detector was 34–360 m/z; the quadrupole temperature was 150 °C, and the acquisition rate was 1 scan/sec.

### 2.3. HPLC Analysis

The analysis of the phenolic compounds was performed according to Longo et al. [34] and Dupas de Matos et al. [35]. Briefly, the separation was carried out on an ODS column (Eurosphere II, C18 stationary phase, 250 × 4.6 mm, 5 µm produce by Knauer) using a Nexera X2 UHPLC (Shimadzu, Milan, Italy) coupled with a UV-Vis diode array detector (Shimadzu). The HPLC flow rate was 0.7 mL min^−1^. The HPLC mobile phase consisted of solvent A (0.1% formic acid in degassed milliQ water) and solvent B (0.1% formic acid in acetonitrile). The gradient method was the following: 0–2.5 min, 99% A; 2.5–50 min 99–75% A; 50–51 min 75–1% A; 51–55 min 1% A; 55–56 min 1–99% A; 56–60 min 99% A.

Phenolic compounds were identified by comparing their chromatographic retention times and UV-Vis spectra with those of pure standard compounds. For this purpose, solutions of standard compounds were injected into the HPLC system. Peaks of different chromatograms were aligned manually. Then, the peak areas of the analytes were integrated using the LabSolutions System software by Shimadzu (Milan, Italy). Calibration curves of pure standard substances (≥98%, Sigma Aldrich, Milan, Italy) were established through the DAD and were used to quantify the concentrations of phenolic compounds. When reference compounds were not available, a calibration with structurally related standard substances was used (gallic acid for protocatechuic acid and syringic acid; caffeic acid for caftaric acid and glutathionyl caftaric acid (GRP); (+)-catechin for (−)-epicatechin) (≥98%, Sigma Aldrich, Milan, Italy). The peak areas were integrated to obtain the concentrations of the identified compounds.

The statistical elaboration of the HPLC data was performed using XLStat (version 2019.2.2.59417, Addinsoft, Paris, France).

### 2.4. Determination of Dissolved O_2_ and SO_2_

A non-invasive optical sensor (L. sensor-700.O2) from an FT system (Alseno, Italy) was used to measure the dissolved oxygen content of wines. This is an IR laser type analyser with a measurement range of 3–21% O_2_ ± 0.3% and an accuracy of ±0.2% O_2_ concentration. Free and total SO_2_ content of wines were analyzed using the Miura One multiparametric analyzer from Exacta + Optech Labcenter (San Prospero, Italy).

### 2.5. Sensory Discrimination Test

A triangle test according to the ISO 4120:2007 method was chosen to evaluate the differences between wines sealed with a ‘conventional’ closure and the ‘blend’ closure.

The sensory panel was formed by 11 assessors (8 males and 3 females, 23 ± 4 years old) who were recruited from among the enology students and technical staff of the Faculty of Science and Technology.

The panelists received 4 sets of 3 wines and were asked to evaluate one set at a time, selecting the odd sample out of each set. Fifty mL of wine were poured into ISO glasses; the glasses were labeled with random three-digit codes. The wine glasses were then presented to the panelists (at a temperature of 18 °C) in a random order, as per the standard methodology for the triangle test. The sensitivity parameters of the test were set at α = 0.5, β = 0.20 and *p*d = 50%. Two sessions were organized due to the relatively low number of assessors.

### 2.6. Multivariate Data Analysis

Multivariate statistical processing of the results was performed to explore the structure of the experimental data using XLStat software (version 2019.2.2.59417, Addinsoft, Paris, France). An alpha value of 0.05 was chosen to determine statistical significance, unless otherwise stated. Regarding the chemical profiles, statistical treatment was performed on the relative abundance of the analyte areas. One-way ANOVA was applied to the data, using closure type as the independent variable, followed by Tukey’s test for post-hoc mean separation. Principal component analysis (PCA) was applied, averaging the same-bottle data (technical repetitions).

## 3. Results

### 3.1. Profile of Volatile Compounds

A total number of 30 volatile compounds were identified in the wines during the 30 months [33] of bottle storage.

Table 2 shows the volatile compounds identified in the four wine samples over 12 months of storage compared to those identified after 30 months. A total of 30 volatile compounds were identified; however, a considerable loss of some of these compounds was observed after long-term storage (30 months). Thirteen compounds (4-ethylbenzoic acid, 2-butyl ester; 1-heptanol; 1-octen-3-ol; hexyl acetate; limonene; 2-ethyl hexanol; methyl benzaldehyde; ethyl benzaldehyde; methyl salicylate; benzenacetic acid, ethyl ester; 2-phenylethylacetate; ethyl dodecanoate; ethyl hexadecanoate) which were initially present in all the wine samples up to twelve months, were not detected at thirty months. On the other hand, four volatile compounds (3-methyl-1-butanol; 2,3-butanediol; 2-hydroxyethyl propanoate; isoamyl lactate) identified at thirty months were not detected in the first twelve months of storage. An interpretation of this observation could be that during storage, some long-chained fresh fruity ethyl esters were hydrolysed to form alcohols and that some esters appeared after 30 months storage—probably due to the condensation of higher alcohols (e.g., isoamyl alcohol) with organic acids formed after fermentation. Additionally, some C7 and C8 higher alcohols and aldehydes disappeared—probably due to oxidation.

The differences in the volatile profile present in wines at 30 months storage were evaluated by one-way ANOVA, taking the closures as the independent factor (Table 3). Four individual one-way ANOVAs were carried out for each wine to test the differences between the ‘blend‘ closure and the ‘conventional‘ one. As a result, only red Lagrein showed a significant difference for 2,3-butanediol (3) and ethyl octanoate (25), which were both higher in red Lagrein wines closed with ‘conventional‘ closures than in those closed with the ‘blend’ closure. No significant difference was found in St. Magdalener, Merlot, nor Lagrein rosé closed with ‘conventional’ or ‘blend’ closures. The volatile compounds that contributed the most to the rosé Lagrein wines sealed with both ‘conventional’ and ‘blend’ closures after 30 months of storage were a diol, a higher alcohol, and 3 esters—which are claimed to give pungent and fruity/floral notes to wines—respectively [51,52]. These compounds were butanediol (3), 1-hexanol (8), ethyl hexanoate (13), ethyl octanoate (25), and ethyl decanoate (28).

The most volatile compounds, such as 3-methyl-1-butanol (2), ethyl butanoate (4), isoamyl lactate (17), octanol (19), and 2-phenylethyl alcohol (21), contributed to the profile of the St. Magdalener wines and are known as compounds that give fresh fruity and floral aromas—excluding 3-methyl-1-butanol and octanol, which give pungent and mushroom/musty notes, respectively [53].

It is also worth noting that octanoic acid (22) and diethyl succinate (23; two volatile compounds associated with fermentations and/or oxidation reactions [33]), and three esters: 2-hydroxyethyl propanoate (5), 2-methylethyl butanoate (6), and 3-methylethyl butanoate (7), distinguished the Merlot samples.

As for the red Lagrein wines, two volatile compounds—acetic acid (1) and 2-phenylethyl alcohol (21)—distinguished the samples.

In summary, there was no differentiation between bottles closed with the two different types of closures after 30 months except for one case—the Lagrein red wine, as highlighted above by the one-way ANOVA. The ANOVA showed no statistically significant differences in terms of the effects of the closure for the Merlot, Lagrein rosé or St. Magdalener.

### 3.2. Non-Volatile Compounds

Gallic acid, protocatechuic acid, caftaric acid, glutathionyl caftaric acid (GRP), caffeic acid, p-coumaric acid, syringic acid, (−)-epicatechin, (+)-catechin and anthocyanins were identified in all four wine samples stored for thirty months (Table 4). Using one-way ANOVA, the differences in the phenolic profile for each type of wine were assessed by considering the closures as a factor. Regarding Merlot wines, GRP and syringic acid showed lower abundances in wines closed with the ‘conventional’ closure (technical cork 1 + 1) than in those closed with the ‘blend’ corks, whereas (+)-cathechin and (−)-epicatechin were higher in the Merlot samples closed with technical 1 + 1 closures.

Concerning the red Lagrein wine, a significant difference was observed in two phenolic compounds: protocatechuic acid and GRP, with a low abundance in Lagrein closed with the ‘conventional’ closure (natural one-piece cork). In rosé Lagrein wine, (−)-epicatechin was higher in rosé closed with the ‘conventional’ closure (technical cork 1 + 1) than in the wine closed with the ‘blend’ closure. Finally, the closures had an effect on the St. Magdalener wine, with a high abundance of caffeic acid, (−)-epicatechin and anthocyanins in the wines closed with the ‘conventional’ closure (agglomerated natural cork), while GRP was low in those closed with the ‘conventional’ closure.

The phenolic compounds that contributed the most to the red Lagrein wines sealed with both ‘conventional’ and ‘blend’ closures after 30 months of storage were p-coumaric and protocatechuic acids, anthocyanins, and caffeic and syringic acids, whereas gallic and caftaric acids, (−)-epicatechin, GRP, and (+)-catechin characterized the Merlot and St. Magdalener samples. Rosé wines had a low content of phenolic compounds due to their particular winemaking process (short maceration time), as has also been reported by Rossetti et al. [33].

### 3.3. Sulfur Dioxide and Dissolved Oxygen Content

One-way ANOVA was performed, considering the closure as a factor, on each wine to understand the influence of the closure type on dissolved oxygen, and free and total sulfites in each wine. The results presented in Table 5 show that only Lagrein rosé closed with the ‘conventional’ closure (technical cork 1+1) had a high content of free SO_2_ (21 mg/L) compared to the ‘blend’ closure (12.5 mg/L). On the other hand, no statistically significant difference was observed in free and total sulfites or dissolved oxygen for the Lagrein red wine, Merlot or St. Magdalener.

### 3.4. Correlations between Sulfur Dioxide, Dissolved Oxygen and Volatile and Phenolic Profiles

A PCA was applied to understand the influence of SO_2_ and dissolved O_2_ content on the wine quality. Figure 1A shows that total SO_2_ and O_2_ contributed together, along with some identified volatile compounds such as butanediol (3), 1-hexanol (8), ethyl hexanoate (13), ethyl octanoate (25), and ethyl decanoate (28), to characterize the Lagrein rosé wines closed with both closures.

Another PCA plot (Figure 1B) was obtained by excluding the rosé Lagrein wines, as they had a lower phenolic content than the other three wines, with the aim of better observing the relationship of SO_2_ and O_2_ with the volatile and phenolic profiles of the red wines. The PCA graph showed that octanoic acid (22) and diethyl succinate (23) showed a stronger relationship with the O_2_ content and mainly characterized the Merlot samples. On the other hand, the content of free and total SO_2_ were directly correlated with (−)-epicatechin.

Two aspects concerning both PCA plots are noteworthy: in both models, (i) glutathionyl caftaric acid was inversely correlated with total SO_2_ content and (ii) three fermentation volatile compounds, including ethyl butanoate (4), isoamyl lactate (17), and octanol (19), were inversely correlated with both free and total SO_2_.

### 3.5. Sensory Evaluation

The triangle test (Table 6) performed on the four wines at 30 months of storage only showed a significant difference for the St. Magdalener and Lagrein rosé wines. Both wines closed with ‘conventional’ closures were different from the ‘blend closure’. On the other hand, no statistical difference was found between the Merlot wines closed with ‘conventional’ and ‘blend’ closures, or for the Lagrein red wines.

## 4. Discussion

For the first time, South Tyrolean wines (three red and one rosé) stored for thirty months in bottles and sealed with different types of closures were studied in terms of changes in volatile and phenolic compounds and their content of free and total sulfites and dissolved oxygen.

The concept of this work was not the absolute quantification of volatile compounds and phenols in these red and rosé wines, but to investigate if storage with different closures could have an impact on the chemical composition of wines. For this purpose, an absolute quantification is not necessary; in fact, ANOVA and multivariate statistical models such as PCA are based on relative abundances (and their variance), and not on an absolute quantification.

One-way ANOVA was considered the best approach for this study, as the “blend” closure was compared for each wine with a conventional closure analyzed after a prolonged storage of the wines in bottles.

The closure type (blend *vs* conventional) did not significantly affect (α < 0.01) free or total SO_2_ or the dissolved oxygen content of any of the wines, with the exception of the free SO_2_ content of Lagrein rosé. This means that, in general terms, all the different types of corks examined mostly ensured good hermeticity during the first 30 months of storage in bottles. In particular, this is important for the winemaker, whose first concern is to maintain the levels of free (and total) sulfites present at bottling, in order to ensure the protection of the wine from oxidation and abnormal fermentations. In the case examined, it was not found that there was a systematic difference in the content of free sulfites in any of the wines according to the type of cork used.

As the differences in free and total sulfite and dissolved oxygen content among the corks used were almost never significant, it is not straightforward to describe the mechanisms that can be predicted from the chemical data here—particularly the influence of oxygen transfer through the cork on various phenolic compounds, sulfur dioxide, and volatiles.

After thirty months of storage, without taking into account the effect of the type of closure, many volatile compounds with a number of carbon atoms from 7 to 18, including long-chain ethyl esters, disappeared, while some esters appeared—probably due to the condensation of higher alcohols (e.g., isoamyl alcohol) with organic acids formed after fermentation. In addition, some higher alcohols and C7 and C8 aldehydes disappeared—probably due to oxidation.

A significant difference in the volatile profiles was observed only in Lagrein red wine closed with the ‘conventional’ closure, which showed a higher content of 2, 3-butanediol and ethyl octanoate compared to the ‘blend’ cork after 30 months. Octanoic acid and diethyl succinate (fermentative volatile compounds) contributed to distinguishing the Merlot samples, which showed the highest dissolved oxygen content. Regarding the differences in phenolic content induced by the closure used, GRP and (−)-epicatechin were the compounds that most consistently changed in the wines. In general, the cinnamic acids, flavanols, and phenolic acids were the most affected by the closure, while anthocyanins differentiated by the closure used only in the St. Magdalener wines. In our case, when the differences were indeed significant, conventional closure-closed wines generally showed higher relative abundances of epicatechin in three of the four wines and lower relative abundances of GRP (in all the three red wines) and other phenolic acids, such as syringic and protocatechuic acids. Finally, anthocyanins and caffeic acid were significantly higher in the St. Magdalener wines closed with conventional closures.

Xing et al. [54] have also reported that storage time has a major effect on the maturation of phenolic compounds, with considerable loss of most phenolic compounds after 18 months in a bottle. Similar results were observed in the present study (30-month storage), where most phenols decreased compared to the results obtained up to 12 months [33]. Xing et al. [54] also demonstrated that there was no strong correlation between the phenolic variation and the dissolved oxygen content. Silva et al. [55] also performed a similar study evaluating the effect of synthetic, agglomerated and natural one-piece cork closures on white, red and rosé wines. Their results showed that the usage of synthetic closures helped to increase the shelf-life of wines within a period of 1–2 years after bottling. Their results are in line with the present research, showing that the different type of closures (a natural one-piece cork, a blend of natural cork microgranules, a technical cork 1 + 1 and an agglomerated natural cork) used to close the four wines (Merlot, Lagrein red and rosé and St. Magdalener) for up to 30 months of storage did not considerably affect the final quality of the wines.

Although the St. Magdalener and Lagrein rosé wines closed with ‘conventional’ or ‘blend’ closures could be discriminated using the triangle test, these differences appeared to be relative, as they did not include all the four wines systematically, but rather appeared to be due to compositional differences in the content of the individual wines. The triangle test was chosen for its ability in aiding the investigation of possible sensory differences that may be present in the wines depending on the type of closure, although the reasons for and the degree of these differences cannot be directly explained with this test.

## Figures and Tables

**Figure 1 foods-10-02918-f001:**
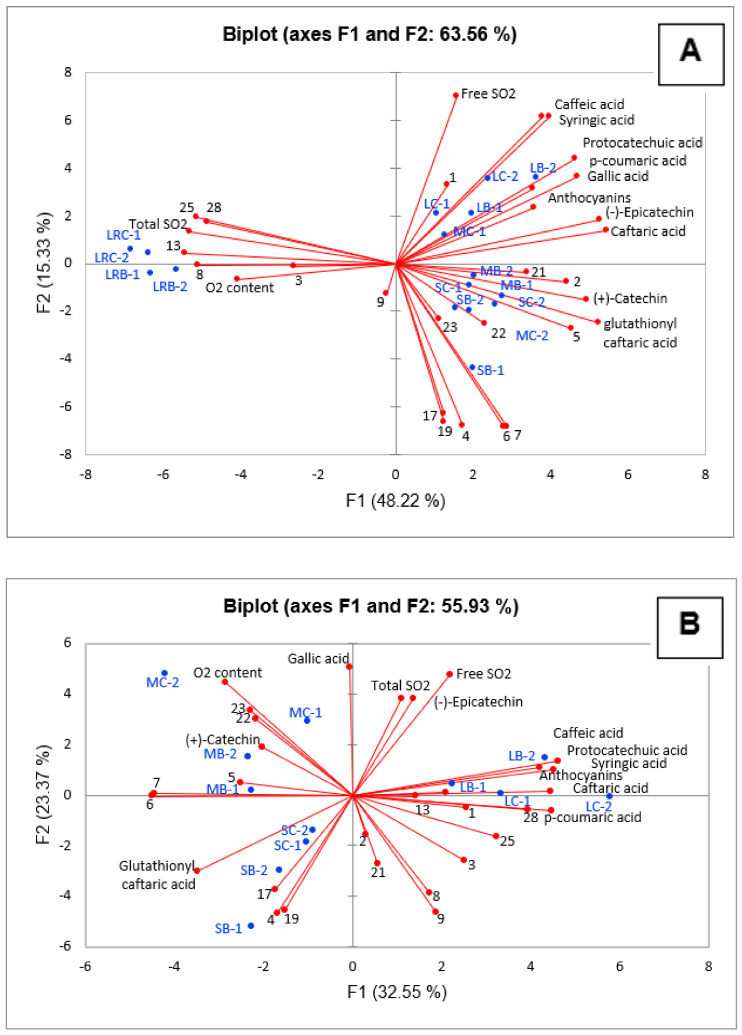
PCA bi-plot of volatile and non-volatile compounds, SO_2_ and dissolved O_2_ contents determined in the wines after 30 months of storage. (**A**) F1 vs F2, Principal Components of three red and one rosé wine; (**B**): F1 vs F2; Principal Components of three red wines excluding the rosé wine. LRB, LRC: Lagrein rosé closed with ’blend’ or ‘conventional’ closure, respectively; SB, SC: same for St. Magdalener; LB, LC: same for Lagrein red; MB, MC: same for Merlot. Volatile compounds are represented by numbers, as listed in Table 2. Values were not averaged for experimental replicates (-1: bottle 1, -2: bottle 2).

**Table 1 foods-10-02918-t001:** The three different ‘conventional’ closures used to close the wine bottles which were compared with a control closure (‘blend’ cork).

Wines	Type of Closures	Images of Closures
Merlot	Blend cork or technical cork (1 + 1)	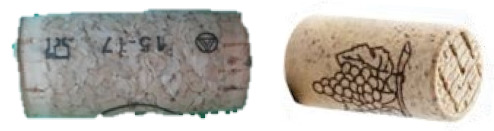 or
Lagrein red	Blend cork or natural one-piece cork	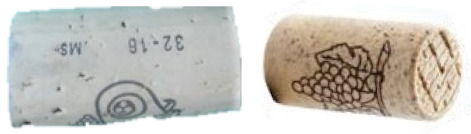 or
Lagrein rosé	Blend cork or technical cork (1 + 1)	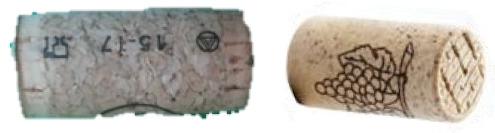 or
St. Magdalener	Blend cork or agglomerated natural cork (0 + 0)	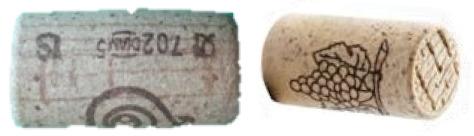 or

**Table 2 foods-10-02918-t002:** Volatile compounds identified during 30 months of storage listed according to their elution order. LRI = Linear retention index.

	Volatile Compounds	Over 12 Months Storage as in [33]	At a 30-Month Storage	LRI (Ref./NIST)	LRI (Measured)	Base Peak (m/z)	Fragmentation Pattern (m/z)
1	Acetic acid	√	√	599 [36]	/	43	43; 45; 60
2	3-methyl-1-butanol	X	√	732 [37]	705	55	42; 55; 70
3	2,3-butanediol	X	√	782 [38]	782	45	45; 57
4	Ethyl butanoate	√	√	803 [36]	772	71	43; 71; 88
5	2-hydroxyethyl propanoate	X	√	793 [39]	792	57	45; 57; 75; 87; 88
6	Ethyl ester of 2-methylbutanoic acid	√	√	846 [36]	824	57	57; 74; 85; 102
7	Ethyl ester of 3-methylbutanoic acid	√	√	859 [40]	827	88	41; 57; 70; 88
8	1-Hexanol	√	√	865 [41]	843	56	56; 69; 84
9	Isopentyl acetate	√	√	876 [41]	850	43	43; 55; 70
10	2-butylester of 4-ethylbenzoic acid	√	X	/	/	133	105; 151
11	1-Heptanol	√	X	969 [41]	/	70	41; 42; 43; 55; 56; 70
12	1-Octen-3-ol	√	X	980 [36]	/	57	43; 55; 57; 72
13	Ethyl hexanoate	√	√	999 [41]	975	88	43; 60; 70; 88; 99
14	Hexyl acetate	√	X	1011 [41]	/	43	43; 55; 56; 61; 69; 84
15	Limonene	√	X	1020 [41]	/	68	67; 68; 93
16	2-Ethyl hexanol	√	X	1028 [42]	/	57	41; 43; 57
17	Isoamyl lactate	X	√	1047 [43]	1042	45	43; 45; 55; 70
18	4-Methyl benzaldehyde	√	X	1076 [44]	/	91	65; 91; 119; 120
19	Octanol	√	√	1070 [41]	1055	56	41; 42; 43; 55; 56; 69; 70; 84
20	4-Ethylbenzaldehyde	√	X	1163 [45]	/	134	91; 105; 133; 134
21	2-Phenylethanol	√	√	1112 [36]	1084	91	65; 91; 122
22	Octanoic acid	√	√	1180 [46]	1152	60	60; 73; 101
23	Diethyl succinate	√	√	1179 [47]	1155	101	45; 55; 73; 101; 129
24	Methyl salicylate	√	X	1192 [41]	/	120	92; 120; 121; 152
25	Ethyl octanoate	√	√	1194 [48]	1169	88	41; 57; 73; 88; 101; 115; 127
26	Benzeneacetic acid, ethyl ester	√	X	1243 [41]	/	91	164; 91;65
27	2-Phenylethylacetate	√	X	1255 [41]	/	104	43; 91; 104
28	Ethyl decanoate	√	√	1403 [49]	1361	88	73; 88; 101; 155
29	Ethyl dodecanoate	√	X	1554 [41]	/	88	43; 73; 88; 101
30	Ethyl hexadecanoate	√	X	1992 [50]	/	88	43; 88; 101

√: detected with internal area >0.01%; X: not detected (internal area <0.01%).

**Table 3 foods-10-02918-t003:** One-way ANOVA of the volatile compounds found in each type of wine closed with ‘conventional’ and ‘blend’ closures (closure effect) at 30 months of storage. Pr > F indicates the *p*-value associated with the F statistic. The asterisk (*) means significantly different using Tukey’s test (α < 0.05). B and C, blend and conventional closure, respectively.

**Merlot**										
Closure	3-Methyl-1-Butanol	2-Hydroxyethyl Propanoate		2-Methyl ethyl Butanoate	3-Methyl ethyl Butanoate	1-Hexanol	Isopenthyl Acetate	Ethyl Hexanoate	Isoamyl Lactate	2-Phenyl Ethanol
C	622245228	40645189		3806540	10057235	15511016	17334871	201290040	3860017	98766115
B	820647919	46158129		5210750	13167523	20750259	39098612	192166261	4403087	175713260
Pr > F	0.341	0.865		0.643	0.746	0.315	0.242	0.804	0.491	0.501
closure	Diethyl succinate	Octanoic acid		Ethyl octanoate	Ethyl decanoate					
C	532233181	128705315		430186222	16189337					
B	249480078	4616469		510187046	31310515					
Pr > F	0.423	0.415		0.862	0.568					
**Lagrein red**										
Closure	3-Methyl-1-butanol	2,3-Butanediol	2-Hydroxyethyl propanoate	1-Hexanol	Isopentyl acetate	Ethyl hexanoate	2-Phenylethanol	Diethyl succinate	Ethyl octanoate	Ethyl decanoate
C	608185318	12511497	24631214	23588648	73852849	241125725	163376742	279803394	987171673	98798491
B	808285867	806359	37373403	22647330	60874527	189831224	129556887	237468679	535496801	42955096
Pr > F	0.691	0.006 *	0.166	0.809	0.214	0.137	0.179	0.223	0.010 *	0.253
**Lagrein rosé**										
Closure	3-Methyl-1-butanol	2,3-Butanediol	1-Hexanol	Isopentyl acetate	Ethyl hexanoate	2-Phenylethanol	Diethyl succinate	Ethyl octanoate	Ethyl decanoate	
C	181321516	7388017	43588910	58215889	540146675	41387280	158422635	2441116143	4952211955	
B	297558079	47935843	45742412	63857184	509214725	70638784	210194281	1614962842	221721242	
Pr > F	0.118	0.487	0.550	0.792	0.436	0.359	0.331	0.065	0.131	
**St. Magdalener**										
Closure	Acetic acid	3-Methyl-1-butanol	2,3-Butanediol	Ethyl butanoate	2-Hydroxyethyl propanoate	2-Methyl ethyl butanoate	3-Methyl ethyl butanoate	1-Hexanol	Isopenthyl acetate	
C	11505643	889501476	2914012	5280037	35636365	4369199	10416494	23082888	83015919	
B	0	754427998	6749973	4663781	39471604	3705953	10245563	24749296	74859273	
Pr > F	0.414	0.533	0.342	0.751	0.574	0.382	0.835	0.234	0.701	
Closure	Ethyl hexanoate	Isoamyl lactate	Octanol	2-Phenyl ethanol	Diethyl succinate	Octanoic acid	Ethyl octanoate	Ethyl decanoate		
C	224036589	5092575	5513034	227373202	354489074	5627357	659384377	22794356		
B	189855333	30786045	16732152	158471891	252616407	4325906	507270573	24993361		
Pr > F	0.151	0.425	0.419	0.223	0.129	0.789	0.143	0.825		

**Table 4 foods-10-02918-t004:** One-way ANOVA of the phenolic compounds found in each type of wine closed with both types of closure (closure effect). Pr > F indicates the *p*-value associated with the F statistic. The asterisk (*) means significantly differences determined by Tukey’s test (α < 0.05). B and C: blend and conventional closure, respectively.

Closure	Gallic Acid	Protocatechuic Acid	Caftaric Acid	Glutathionyl Caftaric Acid	(+)-Catechin	Caffeic Acid	Syringic Acid	(−)-Epicatechin	*p*-Coumaric Acid	Anthocyanins
Merlot										
C	797902	19277	452477	64174	161170	109558	85709	92869	190434	86965
B	775296	20349	442758	68093	143295	96415	91451	80817	179824	50231
Pr > F	0.065	0.063	0.308	0.001 *	0.039 *	0.193	0.045 *	0.003 *	0.371	0.157
Lagrein red										
C	529893	26416	534064	56755	116984	251583	179799	77357	349054	184361
B	547415	29802	554889	61792	117628	244212	212296	100352	378623	133896
Pr > F	0.234	0.026 *	0.459	0.026 *	0.956	0.869	0.068	0.229	0.522	0.278
Lagrein rosé										
C	38832	9592	90022	24781	33893	20066	31342	6048	46235	16893
B	38850	10169	88684	25115	37145	17456	32832	3878	41259	11602
Pr > F	0.966	0.637	0.463	0.668	0.078	0.051	0.102	0.040 *	0.137	0.133
St. Magdalener										
C	144676	17699	471285	66081	173598	135204	83950	82501	245319	171631
B	143419	19867	474859	76162	106769	78370	85387	60521	253125	77102
Pr > F	0.848	0.082	0.748	0.005 *	0.056	0.004 *	0.807	0.001 *	0.250	0.014 *

**Table 5 foods-10-02918-t005:** One-way ANOVA for dissolved oxygen, free and total sulfur dioxide content of wines stored for 30 months and closed with blend closures. Pr > F indicates the *p*-value associated with the F statistic. The asterisk (*) means significantly different groups as determined by Tukey’s test (α < 0.01). B and C: blend and conventional closure, respectively.

Closures	Total Sulfur Dioxide (mg/L)	Free Sulfur Dioxide (mg/L)	Oxygen Content (mg/L)
Merlot			
C	38.50	26.00	1.85
B	26.50	20.00	1.05
Pr > F	0.077	0.238	0.064
Lagrein red			
C	37.00	22.50	0.15
B	28.50	25.00	0.10
Pr > F	0.054	0.649	0.423
Lagrein rosé			
C	96.00	21.00	2.00
B	83.00	12.50	2.00
Pr > F	0.049	0.003 *	1.000
St. Magdalener			
C	36.00	16.00	0.25
B	19.00	11.50	0.05
Pr > F	0.033	0.057	0.106

**Table 6 foods-10-02918-t006:** Triangle test. ns: not significant; * significant (α < 0.05).

Merlot	Lagrein Red	Lagrein Rosé	St. Magdalener
ns	ns	*	*

## Data Availability

The data presented in this study are available on request from the corresponding author.

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
