# Peer review of "Effects of Long-Term Bottle Storage on Red and Rosé Wines Sealed with Different Types of Closures"

_foods, 2021, doi:10.3390/foods10122918_

Round 1
Reviewer 1 Report
This paper studied the effect of different stopper on the total sulfur dioxide content and dissolved oxygen content of wine during long-term storage, as well as the resulting chemical composition difference. Some clarifications/changes should be done.
1 Whether the compounds identified in Table 2 were confirmed by the standard substances.
- Table 3-4. Accurate quantification of volatile and non-volatile compounds is necessary. I suggest adding the accurate quantitative information of the compounds and then recalculating the significance.
- The experimental process and results of the sensory analysis were not fully presented in this paper. It is suggested to combine chemically with sensory for systematic analysis.
Reviewer 2 Report
The authors present a study looking at compositional changes in red and rose wines over a 2.5 year period with different bottle closures. Even though the novelty is very limited and the presentation of the data has flaws, there is potential for this study to be of interest if the authors perform substantial revisions on the manuscript.
Specific comments
Line 2 The title should say "red and rose wines".
Line 18 Lines 18 to 21 seem to be results and should be moved to the appropriate part of the abstract.
line 21 The experimental setup is unclear from this description, especially since "for each wine" and "depending on the wine" in the same sentence is somewhat confusing.
Line 31 The last sentence of this abstract needs to be revised. It is not clear what the authors are trying to say here.
Line 36 I would suggest to use keywords that do not already appear in the title.
Line 48 There are so many definitions of quality ranging from aromatic to chromatic quality. Not all of them are equally affected by bottle aging and it needs to be more specific here.
Line 74 Please add a reference for the influence of humidity.
Line 76 This goes for the whole manuscript. Please revise the use of in-text citations. Those should appear as Author(s), Year, not just the reference number.
Line 109 Did all the corks come from this supplier?
Line 112 The cellar temperature and humidity need to be specified. Also, how much time did the wine spend in the cellar vs. the lab?
Line 120 I would suggest to combine Table 1 and Figure 1. It looks more disconnected than it should.
Line 123 Why did you not use every wine with every cork? It would have made more sense to try every possible comnination.
Line 155 What is the manufacturer or supplier information for the column?
Line 167 What is the manufacturer or supplier information for the column?
Line 168 Manufacturer information?
Line 175 The unit needs to be formatted properly.
Line 179 What was the supplier and purity for all these chemicals?
Line 187 This is confusing. Protocatechuic acid, syringic acid, and epicatechin are available as pure substances. Why did you not use them?
Line 200 Please replace the word "measured" with "analyzed" in some of the sentences to avoid too much repetition.
Line 206 This is an extremely small panel for difference testing. Why only 11 panelists?
Line 245 I assume that the data for 12 months is coming from the previously published study. Please reference that properly here, otherwise in might look like plagiarism.
Line 264 Throughout the manuscript, please replace "superior alcohols" with "higher alcohols".
Line 266 There is most likely a correlation between the oxidation observed here and the SO2 level in the wines.
Line 274 Why is the data not shown? Even though there were no significant differences, the data still has value.
Line 320 Please combine tables 4a to 4d into one table.
Line 361 Please combine tables 5a to 5d into one table.
Line 449 Please provide a data table here.
Line 481 I doubt that this is the only explanation. The lack of differences is probably a results of the small panel size.
Line 482 This discussion is way too short. All analytical data and statistics need to be discussed in great detail. Otherwise, the whole study seems rather insignificant.
Round 2
Reviewer 2 Report
The authors present a revised version of their manuscript that has been improved significantly. I recognize the hard work that went into this and want to thank the authors for making the effort! There are still a few minor corrections that I would suggest.
Specific comments
Line 84 Please double check the format of in-text citations. In this case it should be shown as author, year instead of the number.
Line 113 I would suggest to avoid the term stopper throughout the manuscript since it is not really a technical term and use cork and closure instead.
Line 141 The table looks much better now, however, it looks like the same picture for all types of cork. Please revise.
Line 178 Please mention the complete manufacturer information.
Line 189 Please add the software including version number that you were using.
Line 236 Avoid abbreviations in headers and use oxygen and sulfur dioxide instead.
Line 283 I am still not entirely happy with the reference of the earlier publication here. For me, it makes sense to re-evaluate after 30 months and compare to the 12 months results, however, Table 2 lists the results side by side without the proper reference. There must be a better way to do this.
Line 423 It looks like there is only one Table 5 now but the text still references different tables.
Line 564 I would like to see a little more discussion here. Please explain some of the mechanisms that you can predict from your data, especially the influence of oxygen transfer through the cork on different phenolic compounds, sulfur dioxide, and selected volatiles. There are many things happening involving an oxidation cascade with hydrogen peroxide that includes all the compounds you analyzed. There is much more that you can extract from your data.
Author Response
Please see attached file containing a point-by-point response to the reviewer’s comments.

This manuscript is a resubmission of an earlier submission. The following is a list of the peer review reports and author responses from that submission.
Round 1
Reviewer 1 Report
The paper by Betnga et al. described the impact of different types of stoppers on wine quality after a storage period of 30 months.
Although the issue of the submitted paper is interesting to the wine producers, the main conclusion was already known from the previous paper (doi.org/10.3390/molecules25184276). Also, in the results and discussion parts, the authors only compared the obtained results without a deep insight into the influencing mechanism.
Besides, it is not acceptable that Tables 2a-b and 3a-e do not shown the values, but only the letters that indicate the existence of statistically significant differences. On the other hand, in Tables 4a-e it is necessary to show standard deviations.
The list of references contains literature number 1 twice, so it is necessary to correct all the numbers under which the references are listed.
My suggestion is that the paper should not be accepted for publication in this form. Eventually, after adequate rearrangement and shortening, the work could be re-submmited as a short communication.
Reviewer 2 Report
To authors:
The study is interesting and present valuable data but needs more focused in the initial profile in each wine. In general, there are some issues that need revision.
Material and methods section
Lines 394-396. It is necessary to state the vinification protocol in each type of grape variety.
Lines 402-404. The authors say “… for most of the storage time…” but it is necessary to state what time of the total time is the “most” and especially says the storage conditions in the cellar, i.e, temperature and humidity. Also, how much was the time in laboratory conditions? and what is the “medium relative humidity?”
Lines 406-407. Why the authors only use two replicates? Most of the assays use as minimum three replicates and more than this. The number of replicates is important for a correct statistical analysis.
Line 471. In the sensory discrimination test, it is necessary say the laboratory condition, for example, temperature, relative humidity and especially the wine temperature, as well as the wine sample volume served to each panelist. For other part, this sensory analysis is only discriminative and it does not serve to appreciate the quality of wine.
Results section
Lines 104-107. In my opinion, the wine samples need an initial analysis of phenolic compounds and volatiles and specially SO2. This first analysis explained the differences after 30 months of storage. The authors only present the final data but is necessary the initial profile in each wine.
For other hand, why the author establishes differences between wines? These wines are different because are made of different grape varieties. In my opinion is a mistake the comparison between a Merlot wine and a rose wine. The vinification process is very different, in term of the presence of skins and seeds in red wine vinification compared with rose wines, so, these differences explain the presence or absence of certain compounds, that is the differences are due to the different vinification process and not to the treatments applied in this assay. This matter affects all the analysis published in this experiment. Please clarify as possible.
Reviewer 3 Report
Overview:
The present study evaluated the effects of different stoppers on the chemical profile of red and rose wines during long-term storage.
Overall, I believe that this work is well prepared. The introduction provides sufficient background and presents the objectives in a clear manner. The methods are carefully described.
Given the assessment of volatile and non-volatile chemical profiles, the free and total SO2 and the dissolved O2 content over 30 months of storage in bottle, the current study informs on the effects of bottle stoppers on wine quality. The topic of the paper is relevant and of interest for the readers of the journal.
However, some issues should be addressed.
Major comment:
Section 3. Discussion: The results are well presented, but the discussion of many pats needs to be strengthened. Moreover, some discussion elements are together with the related result, and others are gathered together in lines 364-391; this makes the Results section quite extensive in comparison with the Discussion section, which lacks in length.
Also, in this context, despite the fact that the ‘Conclusions’ section is not mandatory (as highlighted in the Instructions for authors) I feel that the Discussion section was somewhat used as a means to summarize the findings.
I suggest the authors decide on a single approach in presenting the results and discussion.
Minor comment:
Materials and methods, Section 4.1.: The design of the study is clearly stated. Fourteen wine samples seem sufficient for the outcome to be adequately interpreted. Even though a comparison of all 4 types of stoppers for each wine would have been more insightful, I believe that the findings of the present study are representative enough for the employed design.